# MADAR: Efficient Continual Learning for Malware Analysis with Diversity-Aware Replay

## Abstract

Millions of new pieces of malware are introduced each year. This poses significant challenges for antivirus vendors, who use machine learning to detect and analyze malware, and must keep up with changes in the distribution while retaining knowledge of older variants. Continual learning (CL) holds the potential to address this challenge by reducing the storage and computational costs of regularly retraining over all the collected data. Prior work, however, shows that CL techniques designed primarily for computer vision tasks fare poorly when applied to malware classification. To address these issues, we begin with an exploratory analysis of a typical malware dataset, which reveals that malware families are diverse and difficult to characterize, requiring a wide variety of samples to learn a robust representation. Based on these findings, we propose $\underline{M}$alware $\underline{A}$nalysis with $\underline{D}$iversity-$\underline{A}$ware $\underline{R}$eplay (MADAR), a CL framework that accounts for the unique properties and challenges of the malware data distribution. We extensively evaluate these techniques using both Windows and Android malware, showing that MADAR significantly outperforms prior work. This highlights the importance of understanding domain characteristics when designing CL techniques and demonstrates a path forward for the malware classification domain.

## 1 Introduction

Advances in machine learning have significantly improved detection and classification of malicious software, with notable success across various settings, such as Windows executables (Dahl et al., 2013; Kovacs, 2018), PDFs (Maiorca et al., 2012), and Android applications (Arp et al., 2014). Traditional models, trained on static datasets, are expected to perform well on new data under the assumption of a constant data distribution. In reality, though, both malicious (i.e., *malware*) and benign software (i.e., *goodware*) are ever-evolving and require regular model updates to keep up with these changes in data distribution to maintain effectiveness. For example, the AV-TEST Institute logs about 450,000 new malware samples daily (AV-TEST, 2023), and VirusTotal processes about one million new submissions each day (VirusTotal, 2023).

Training a malware classification model solely on new data can lead to *catastrophic forgetting (CF)* (French, 1999), which may result in both misclassifying goodware and allowing attackers to bypass detection with older malware strains. To address this, antivirus vendors can keep older samples and retrain over all of them during model updates, but the enormous volume of these samples makes the storage and computational costs of this approach excessive. Continual learning (CL) offers a solution to this problem by enabling models to adapt to new data without the need for maintaining large datasets or extensive retraining (van de Ven et al., 2020; Bhat et al., 2023).

While designs for CL have been extensively studied in the context of computer vision (Shin et al., 2017; Hsu et al., 2018; van de Ven et al., 2020), there are very few such studies in the malware classification domain. An exception is the work of Rahman et al. (2022), who found that none of the CL techniques originally designed for computer vision problems offer acceptable performance in malware classification, due in part to the strong semantics of malware features and the high level of diversity found in the malware ecosystem.

In this study, we first delve into the complexities of malware data distributions using the EMBER dataset (Anderson & Roth, 2018) of Windows malware and goodware. Our analysis highlights the diversity in malware, both between and even within *families*—groups of related malware. Leveraging this insight, we devise MADAR, $\underline{M}$alware $\underline{A}$nalysis with $\underline{D}$iversity-$\underline{A}$ware $\underline{R}$eplay, a replay-based continual learning strategy that accounts for diversity and achieves improved malware classification performance. MADAR replays a mix of representative samples and novel samples (i.e., outliers) to enhance the model's ability to retain knowledge and identify new malware variants despite memory constraints. Our techniques use Isolation Forests (IF) to find these novel samples.

We then evaluate MADAR with comprehensive experiments on the EMBER dataset in three CL scenarios that mirror common malware classification tasks: domain incremental learning (Domain-IL), class incremental learning (Class-IL), and task incremental learning (Task-IL). Additionally, we have curated two new benchmarks of Android malware from the AndroZoo repository (Allix et al., 2016): *AZ-Domain* for Domain-IL experiments and *AZ-Class* for Class-IL and Task-IL scenarios. Our results on these datasets confirm that MADAR is indeed effective and much better than prior state-of-the-art CL methods in the face of realistic data distribution shifts.

In summary, the contributions of this study are:

- We provide an exploratory analysis of the diversity of malware distributions and show how it creates unique challenges for continuous learning.
- We develop two large-scale, realistic Android malware benchmarks covering all three CL scenarios – Domain-IL, Class-IL, and Task-IL.
- In Domain-IL scenarios, we show that MADAR performs much better than prior CL techniques. On the AZ dataset, for example, MADAR comes within 0.4% average accuracy of the joint training baseline using just 100K training samples versus 680K for joint.
- MADAR is also effective in Class-IL scenarios, where it consistently outperforms all prior methods over a wide range of budgets. With a budget of 20K training samples on EMBER, MADAR gets an average accuracy of $85.8\%$ versus $66.8\%$ for the best prior method.
- For Task-IL, MADAR outperforms all prior methods across all memory budgets for both the EMBER and AZ datasets. For example, in the AZ dataset, the MADAR-U variant achieves an average accuracy of 98.7% (within 0.1% of joint) with a budget of only 20K replay samples (versus approximately 250K for joint).

## 2 RELATED WORK

**Replay in Continual Learning.** The fundamental challenge in developing a CL system is addressing catastrophic forgetting (CF), and one of the widely studied methods to overcome CF is *replay* (Zhang et al., 2024; Elsayed & Mahmood, 2024; Bhat et al., 2023; Rebuffi et al., 2017). These techniques can further be classified into one of two major subcategories – exact replay and generative replay.

Exact-replay techniques are designed to choose replay samples from previously learned data to be combined with new samples for retraining. The goal of these techniques is to maximize the performance with minimal replay samples (Rolnick et al., 2019; Chaudhry et al., 2019a; Rebuffi et al., 2017; Bhat et al., 2023). Selecting and using the replay samples involves determining a memory budget, denoted as $\mathcal{M}$, and choosing an optimal value of $\mathcal{M}$ remains an open research question (Aljundi et al., 2019; Chaudhry et al., 2019b). Generative or pseudo-replay strategies are designed to replicate the original data (Li & Hoiem, 2017; Shin et al., 2017; van de Ven et al., 2020). These techniques either generate a representative of the original data using a separate generative model or generate pseudo-data by using an earlier model's predictions as soft labels for training subsequent models. We experimentally compare the performance of MADAR against the leading exact and generative replay techniques, and we find that it's diversity-aware approach outperforms all of them in the malware domain.

**CL in Malware and Related Domains.** Despite extensive work in CL, very few studies have studied CL in the malware domain. Rahman et al. (2022) were the first to explore CL for malware classification. They concluded that existing CL methods fall short in tackling CF in malware classification systems due to differences in the underlying nature of the data distribution shifts that occur

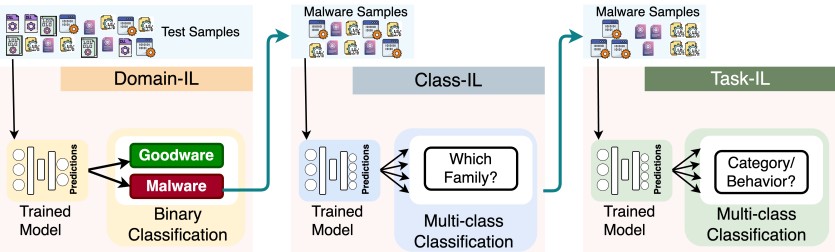

Figure 1: CL scenarios in a typical malware analysis pipeline.

in practice versus those explored in the computer vision domain. Malware representations use tabular features with strong semantic constraints that limit the space of feasible samples, and within that space, samples exhibit a high degree of diversity. Replay-based techniques were found to perform better compared to other approaches in this setting, leading to our focus on these methods.

Related strains of work address *online learning* (Xu et al., 2019) and *concept drift* (Chen et al., 2023) in malware classification. These works do not address overcoming CF.

Another CL domain in cybersecurity is network intrusion detection (NID), looking for malicious activity based on network packets. Channappayya et al. (2024) explored a replay-based CL technique that incorporates class-balancing reservoir sampling and perturbation assistance for parameter approximation NID. Another recent work explored semi-supervised CL for NID in a class incremental setting (Amalapuram et al., 2024). We note that NID is a different domain with different data characteristics than malware. Further, these works do not focus on reducing CF.

## 3 CL Scenarios in the Malware Analysis Pipeline

Continual Learning (CL) is categorized into three scenarios: Domain Incremental Learning (Domain-IL), Class Incremental Learning (Class-IL), and Task Incremental Learning (Task-IL) (van de Ven et al., 2022). In this section, we explain how the three CL scenarios fit into a typical malware analysis pipeline (see Figure 1).

**Domain-IL.** The first step in the pipeline is the binary classification problem of distinguishing between goodware and malware. Each day, VirusTotal receives one million never-before-seen samples (VirusTotal, 2023), highlighting the persistent and evolving nature of software.This underscores the importance of rapidly integrating these new samples into operational systems to protect against evolving threats. In addition, with the continual emergence of new benign software programs and the massive class imbalance in practice (i.e., significantly more goodware than malware), it is of utmost importance to not increase the false positive rate of the classifiers.

In this adversarial context, attackers may deploy older malware to evade detection by systems that have *forgotten* their patterns, necessitating a balance between adapting to new threats and preventing CF. To evaluate CL schemes in this regard, we segment our Domain-IL datasets into blocks of time based on the date of each sample.

**Class-IL.** Once software is determined to be malicious, the next task in malware analysis involves classifying malware into families—groups of programs with substantial code overlap and similar functionality, as recognized by experts (Zhu et al., 2020). For instance, the `zeus` banking trojan has evolved into 556 variants across 35 families, including `citadel` and `gameover`. Defining a new class relies on consensus from multiple anti-virus engines and occurs when a significant set of similar samples forms a new family (Kantchelian et al., 2015; Zhu et al., 2020). In our incremental multi-class model, we start with a set of known malware families and add new ones as they emerge, continuously adjusting and assessing the model across all known classes.

**Task-IL.** In malware analysis, leveraging insights from additional methods can prove beneficial. This may involve identifying the broader malware category (e.g., adware, ransomware, etc.), malware behaviors (Berlin et al., 2015), or the infection vector (e.g., phishing, downloader, etc.). Task-IL encapsulates this concept of constrained tasks, where the introduction of a new task may sym-

bolize a new category or behavior set. This event occurs less frequently than adding a new family, as found in Class-IL. Unlike Class-IL, the task identity is provided to the model at test time, significantly simplifying the problem. In malware, this could mean learning the task identity from a separate model, manual analysis, or field reports of the malware's behavior. As our datasets do not possess naturally defined tasks, we partition our dataset into tasks comprising an equal number of independent and non-overlapping classes to act as a proxy to new behaviors, following common practice in the CL literature (van de Ven et al., 2022; 2020). In other words, a given task would be to perform family classification among one subset of families, and the subset that each sample belongs to is known to the classifier. The model is expected to be able to handle multiple tasks at once, and new tasks are being added during each experiment.

## 4 MADAR: Malware Analysis with Diversity-Aware Replay

### 4.1 Exploratory Analysis of the EMBER Dataset

In this section, we provide an analysis of the EMBER dataset (Anderson & Roth, 2018), a widely used benchmark for Windows malware classification. The analysis sheds light on the data distribution across various families and tasks, aiding in selecting representative samples for replay. We identified 2,899 unique malware families within a subset of EMBER, and an additional 11,433 samples lacking clear family labels were assigned the label *Other*.

We investigate the prevalence of malware families over time, distinguishing between recurring and newly identified families each month. Unlike many datasets used in CL research, we see significant churn in the representation of families over time. Of the 913 families seen in January, for example, only 551 are seen in February, while 425 new families emerge. This churn indicates a potential issue in training data continuity, which may aggravate catastrophic forgetting and underscores the need for different CL strategies in the malware domain. Generally, each family has its own distribution pattern, and together these patterns make up the total distribution of malware for a particular month.

Worse, many malware samples do not have family labels at all. Correctly labeling samples is challenging, requiring time and expert knowledge (Kantchelian et al., 2015), so the lack of labels matches real-world conditions. Family labels for malware are based on the *av-class* labels provided by the av-test engine (AV-TEST, 2023). The *Other*-labeled samples do not seem from our analysis to align with other families, meaning that many of them indeed come from unknown families.

Furthermore, the prominent malware families change with the evolution of tasks. The 10 most common families vary greatly across tasks. For example, the `emotet` malware family was the most consistent, appearing in 11 out of 12 tasks. The next most consistent families were `fareit` and `zusy`, appearing in eight and seven tasks, respectively. This indicates considerable concept drift in malware data, highlighting the need to regularly update classifiers.

Challenges in the malware domain also arise from its complexity. Many malware families display complex distributional patterns in feature space, making for additional diversity within classes. Figure 2 shows a t-SNE projection of EMBER features for all malware samples from January 2018. Each family (represented by color) is not clustered into a single well-defined region. Rather, the larger families are split up and spread out in feature space. To accurately represent the malware distribution, it is thus important to

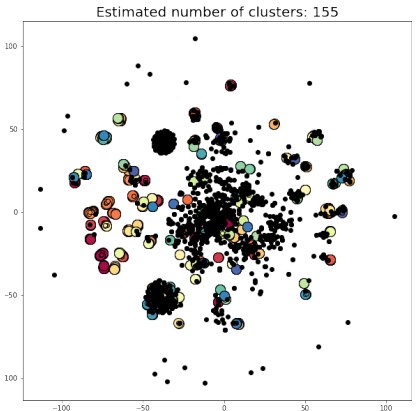

Figure 2: t-SNE projection of EMBER malware, Jan. 2018

select samples not only from each family, but also from multiple areas within each family. This may explain why prior CL techniques designed for computer vision datasets are less effective when applied to the malware domain (Rahman et al., 2022).

In light of these results, we propose that selecting replay samples based on families and variations within families could more effectively capture the diversity within the data and help mitigate CF.

---

**Algorithm 1:** MADAR in Domain-IL

---

**Input** : $c$ – Task, $X_c, Y_c$ – Samples and labels, $\mathcal{P}$ – Data pool, $\beta$ – Memory budget, $\gamma$ – Mal/goodware split, $\Omega$ – anomalous/similar split, $\xi$ – Ratio budgeting, $\Psi$ – Uniform budgeting

1 **init** $\mathcal{P}$; **init** $\mathcal{D} \leftarrow \{M_f : M_c\}$;
2 **if** $c = 0$ **then**
3     $\mathcal{P} \leftarrow X_c, Y_c; X_{good}, X_{mal} \leftarrow \mathcal{P}; \nabla \mathcal{D} \leftarrow X_{mal}; X_{train}, Y_{train} \leftarrow X_c, Y_c;$
4 **else**
5     $X_{good}, X_{mal} \leftarrow \mathcal{P}; \beta_M, \beta_G \leftarrow \beta \cdot \gamma; \beta_A, \beta_S \leftarrow \beta_M \cdot \Omega;$
6     **if** $\Psi$ **then**
7         $\mathcal{F} \leftarrow \mathcal{D}; \mathcal{B}_{\mathcal{F}} \leftarrow \beta_M / \mathcal{F};$
8     $R_{mal} \leftarrow [\,]; $ **for** $X_f \subseteq X_{mal}$ **do**
9         $\mathcal{F}_{\mathcal{MC}} \leftarrow X_f; $ **if** $\xi$ **then**
10             $\mathcal{MC} \leftarrow \mathcal{D}; \mathcal{B}_{\mathcal{F}} \leftarrow (\mathcal{F}_{\mathcal{MC}} / \mathcal{MC}) \cdot \beta_M;$
11         **if** $\mathcal{F}_{\mathcal{MC}} \leq \mathcal{B}_{\mathcal{F}}$ **then**
12             $R_{mal}.\text{append}(X_f);$
13         **else**
14             $(A_f, S_f) \leftarrow \text{IF}(X_f, \beta_A, \beta_S); R_{mal}.\text{append}(A_f, S_f);$
15     $R_{good} \leftarrow \text{sample}(X_{good}, \text{len}(R_{mal})); X_{replay} \leftarrow (R_{good}, R_{mal});$
16     $Y_{replay} \leftarrow ([0] \times \text{len}(R_{good}), [1] \times \text{len}(R_{mal}));$
17     $X_{train} \leftarrow \text{concat}(X_c, X_{replay}); Y_{train} \leftarrow \text{concat}(Y_c, Y_{replay});$
18     $\mathcal{P}.\text{append}(X_c, Y_c); \nabla \mathcal{D} \leftarrow X_{mal};$
19 **return** $(X_{train}, Y_{train})$

---

## 4.2 MADAR

Here, we introduce the MADAR framework for CL in malware classification that uses a diversity-aware replay buffer to account for the diversity of samples we saw in EMBER.

Building on our analysis in Section 4.1, we postulate that stratified sampling—where replay samples are chosen based on their representation in malware families—may better preserve the model's stability compared with random sampling as used in global reservoir sampling (GRS) (Vitter, 1985; Rahman et al., 2022). Moreover, we also seek to capture the diversity *within* each family's data distribution. Let $\mathcal{D}_f = \{x_1, x_2, \ldots, x_n\}$ represent the data samples belonging to a specific family $f$. We define two subsets of interest within this data: the *representative samples* $\mathcal{S}_f \subset \mathcal{D}_f$ capture the frequently occurring samples within the family and the *anomalous samples* $\mathcal{A}_f \subset \mathcal{D}_f$ capture rare samples. The selected set of replay samples $\mathcal{S} = \mathcal{S}_f \cup \mathcal{A}_f$ captures the diversity within the family's data distribution by balancing representative and anomalous samples. While any single anomalous sample is not as important to learn and remember as a single representative sample, a collection of anomalous samples helps to track the diversity within a class.

*Isolation Forest (IF)* (Liu et al., 2008) is a technique for identifying outliers in high-dimensional data. IF uses decision trees to isolate anomalous data points based on the intuition that they are easy to separate from the rest of the data. An important parameter in IF is the contamination rate $C_r$, which represents the expected fraction of outliers in the data. We found that $C_r = 0.1$ works best and used it in all our experiments. The algorithm for MADAR in the Domain-IL setting is provided in Algorithm 1. The algorithms for Class-IL and Task-IL are presented in Appendix A.

### 4.2.1 PROCEDURE

We now describe MADAR using the framework of Domain-IL; the process is similar for Class-IL and Task-IL. The procedure begins by initializing a global data pool $\mathcal{P}$, containing both goodware and malware samples, and a dictionary $\mathcal{D}$ that tracks malware families and their frequencies in the data up to the current task.

For each new task $c$, MADAR divides the data into goodware ($X_{good}$) and malware ($X_{mal}$) subsets from $\mathcal{P}$, allocating memory budgets $\beta_M$ for malware and $\beta_G$ for goodware from the total memory budget $\beta$, based on a split ratio $\gamma$:

$$\beta_G = \gamma \cdot \beta, \quad \beta_M = (1 - \gamma) \cdot \beta.$$

For balanced datasets like EMBER, $\gamma = 0.5$ ensures an equal split between malware and goodware. For an imbalanced dataset, it is better to tune $\gamma$. Our Android malware (AZ) datasets, for example, have a 9:1 ratio of goodware to malware, so we use $\gamma = 0.9$.

Before training for a new task, MADAR incrementally trains the classifier using a combination of new samples from the current task and replay samples from previous tasks. The replay samples include both goodware ($R_{good} \subset X_{good}$) and malware ($R_{mal} \subset X_{mal}$), with $R_{mal}$ sampled from specific malware families rather than randomly from all of $X_{mal}$.

For each family $f$, we set its *family budget* $\mathcal{B}_f$— the number of samples to select from $f$—using two sub-sampling variants: *Ratio budgeting* and *Uniform budgeting*.

- **Ratio Budgeting:** Select the number of samples from a family $f$ proportional to that family's representation in the dataset. The family budget $\mathcal{B}_f$ is $\mathcal{B}_f = \frac{|X_f|}{|X_{mal}|} \cdot \beta_M$, where $|X_f|$ is the number of samples in family $f$, and $|X_{mal}|$ is the total number of malware samples. This strategy may be more suitable in binary classification, as it provides proportional representation of the malware families for training on the malicious class.
- **Uniform Budgeting:** In this method, the memory budget $\beta_M$ is uniformly distributed across all malware families: $\mathcal{B}_f = \frac{\beta_M}{|\mathcal{F}|}$, where $|\mathcal{F}|$ is the total number of malware families. Compared with Ratio budgeting, Uniform budgeting may work well for multi-class classification to determine which family a sample belongs to, as it ensures better class balance.

Within each family $f$, we further split the family budget $\mathcal{B}_f$ into two parts: representative samples $S_f$ and anomalous samples $A_f$, using IF, controlled by a split parameter $\alpha$:

$$|S_f| = \alpha \cdot \mathcal{B}_f, \quad |A_f| = (1 - \alpha) \cdot \mathcal{B}_f$$

We found empirically that a balanced split ($\alpha = 0.5$) between representative and anomalous samples provides optimal performance. In this setup, the model learns equally the core class characteristics from representative samples and less common variations from anomalous samples.

The malware replay set $R_{mal}$ is then constructed by combining the representative and anomalous samples from all malware families:

$$R_{mal} = \bigcup_{f \in F} \{S_f \cup A_f\}.$$

The total replay set consists of both goodware and malware replay samples, which are then concatenated with the new task samples to form the training set for the current task $c$. After training, the data pool $\mathcal{P}$ is updated with the new task samples, $\mathcal{P} \leftarrow \mathcal{P} \cup \{X_c, Y_c\}$, and the malware family dictionary $\mathcal{D}$ is updated to reflect the new frequencies of malware families in $X_{mal}$: $\mathcal{D} \leftarrow \mathcal{D} + freq(X_{mal})$.

## 5 EVALUATION

### 5.1 EXPERIMENTAL SETUP, DATASETS, AND BASELINES

We now present the results of our MADAR framework in the Domain-IL, Class-IL, and Task-IL scenarios for EMBER and AZ datasets. We use the following two abbreviations to denote our techniques—**MADAR-R** for MADAR-Ratio and **MADAR-U** for MADAR-Uniform. For all three scenarios, we compare MADAR with the most widely studied replay-based CL techniques: experience replay (ER) (Rolnick et al., 2019), average gradient episodic memory (AGEM) (Chaudhry et al., 2019a), deep generative replay (GR) (Shin et al., 2017), Replay-through-Feedback (RtF) (van de Ven & Tolias, 2018), and Brain-inspired Replay (BI-R) (van de Ven et al., 2020). In addition, we compare MADAR with iCaRL (Rebuffi et al., 2017), a replay-based technique specifically designed for Class-IL. Furthermore, we compare MADAR with Task-specific Attention Modules in Lifelong learning (TAMiL) (Bhat et al., 2023) which is designed for Class-IL and Task-IL scenarios. We observe that recent works mostly focus on Class-IL and Task-IL scenarios which limits what we can compare with in the Domain-IL scenario. The results of the best-performing method, as well as those within the error range of the best results, are highlighted in the results tables. We built upon the code of the prior work by Rahman et al. (2022).

Table 1: **Domain-IL**: Global average accuracies.

| Group | Method | EMBER | | | | AZ | | | |
| | | Budget | | | | Budget | | | |
| | | 1K | 100K | 200K | 400K | 1K | 100K | 200K | 400K |
|---|---|---|---|---|---|---|---|---|---|
| Baselines | Joint | 96.4±0.3 | | | | 97.3±0.1 | | | |
| | None | 93.1±0.1 | | | | 94.4±0.1 | | | |
| Prior Work | ER | 80.6±0.1 | 69.9±0.1 | 70.0±0.1 | 70.0±0.1 | 40.4±0.1 | 42.6±0.1 | 44.0±0.1 | 48.6±1.1 |
| | AGEM | 80.5±0.1 | 70.0±0.1 | 70.0±0.2 | 70.0±0.1 | 45.4±0.1 | 53.7±0.6 | 54.2±0.3 | 56.7±0.3 |
| | GR | 93.1±0.2 | | | | 93.3±0.4 | | | |
| | RtF | 93.2±0.2 | | | | 93.4±0.2 | | | |
| | BI-R | 93.4±0.1 | | | | 93.5±0.1 | | | |
| | GRS | **93.6±0.3** | **95.3±0.7** | **95.9±0.1** | **96.0±0.3** | 95.3±0.1 | **97.1±0.1** | **97.1±0.1** | **97.2±0.1** |
| Ours | MADAR-R | **93.7±0.1** | **95.3±0.6** | **96.0±0.1** | **96.1±0.1** | **95.8±0.1** | 97.0±0.1 | 97.0±0.1 | 97.0±0.1 |
| | MADAR-U | **93.6±0.2** | **95.3±0.1** | 95.5±0.1 | 95.8±0.1 | **95.7±0.1** | 95.2±0.1 | 95.4±0.1 | 96.3±0.2 |

In this study, we utilize large-scale malware datasets, including the EMBER dataset (Anderson & Roth, 2018), a widely used benchmark for Windows malware classification, and two Android malware datasets derived from AndroZoo (Allix et al., 2016), specifically assembled for this research. We compare our approach against two baselines: *None*, where the model is trained sequentially on each new task without any CL techniques, serving as an informal lower bound; and *Joint*, which trains on both new and previous data at each step, representing an informal upper bound. Although resource-intensive, *Joint* ensures consistently strong results. Additionally, we introduce Global Reservoir Sampling (GRS), an approach based on *reservoir sampling* (Vitter, 1985) and Rahman et al. (2022), which provides an unbiased representation of class distributions and serves as a strong point of comparison for our diversity-aware approach.

More details on the experimental setup and additional results with more memory buffer configurations are provided in Appendix B and Appendix C, respectively.

## 5.2 DOMAIN-IL

In EMBER, we have 12 tasks, each representing the monthly data distribution spanning January–December 2018. Our results, detailed in Table 1, present a nuanced view of each method's performance, reported as the average accuracy over all tasks $\overline{\mathbf{AP}}$. The informal lower and upper performance bounds for this configuration can be approximated by the *None* and *Joint* methods, which get $\overline{\mathbf{AP}}$ of 93.1% and 96.4%, respectively. Meanwhile, *GRS* represents a strong baseline for unbiased sampling without awareness of sample diversity.

At lower budget of 1K, GRS, MADAR-R, and MADAR-U exhibit competitive performance, all significantly better than prior work with $\overline{\mathbf{AP}}$ above 93.6%, indicating their effective utilization of limited resources. ER and AGEM performed far below even the *None* baseline, while GR could only match it. For higher budgets, GRS and MADAR methods all show excellent performance. At a 200K budget, MADAR-R yields $\overline{\mathbf{AP}}$ of 96.0%, close to the 96.4% reached by the Joint baseline that used over 670K samples. GRS is competitive, while Uniform strategies are only slightly behind.

For the experiments with AZ-Domain, we have 9 tasks, each representing a year from 2008 to 2016. The performance of each method is shown in Table 1 as $\overline{\mathbf{AP}}$ and compared with two baselines: *None* at $94.4 \pm 0.1$ and *Joint* at $97.3 \pm 0.1$.

As with EMBER, we find that our MADAR techniques greatly surpass previous methods like ER, AGEM, GR, RtF, and BI-R for every budget level. For lower budgets like 1K, MADAR-R slightly outperforms GRS and is within 1.5% of *Joint*. For higher budgets (100K-400K), MADAR-R perform well – in line with GRS and just slightly below *Joint*, which requires 680K training samples.

In summary, our results empirically depict the effectiveness of MADAR's diversity-aware sample selection in maximizing the efficiency and effectiveness of a malware classifier in Domain-IL. MADAR-R is either better or on par with GRS and significantly better than prior work.

Table 2: **Class-IL**: Global average accuracies.

| Group | Method | EMBER Budget | | | | AZ Budget | | | |
|---|---|---|---|---|---|---|---|---|---|
| | | 100 | 1K | 10K | 20K | 100 | 1K | 10K | 20K |
| Baselines | Joint | 86.5±0.4 | | | | 94.2±0.1 | | | |
| | None | 26.5±0.2 | | | | 26.4±0.2 | | | |
| Prior Work | TAMiL | 32.2±0.3 | 35.3±0.2 | 38.2±0.3 | 38.8±0.2 | 53.4±0.3 | 57.6±0.3 | 63.5±0.1 | 67.7±0.3 |
| | iCaRL | 53.9±0.7 | 60.0±1.0 | 64.6±0.8 | 66.8±1.1 | 43.6±1.2 | 61.7±0.7 | 81.5±0.6 | 84.6±0.5 |
| | ER | 27.5±0.1 | 28.0±0.1 | 28.0±0.1 | 28.2±0.1 | 50.8±0.7 | 58.9±0.2 | 62.9±0.7 | 64.2±0.4 |
| | AGEM | 27.3±0.1 | 27.7±0.1 | 28.2±0.1 | 28.2±0.1 | 27.3±0.7 | 27.1±0.3 | 28.2±1.0 | 28.0±0.8 |
| | GR | 26.8±0.2 | | | | 22.7±0.3 | | | |
| | RtF | 26.5±0.1 | | | | 22.9±0.3 | | | |
| | BI-R | 26.9±0.1 | | | | 23.4±0.2 | | | |
| | GRS | 51.9±0.4 | 75.4±0.7 | 83.5±0.1 | 84.6±0.2 | 43.8±0.7 | 70.2±0.4 | 86.4±0.2 | 89.1±0.2 |
| Ours | MADAR-R | **68.0±0.4** | 76.0±0.3 | 83.2±0.2 | 84.0±0.2 | **59.4±0.6** | 71.9±0.5 | 86.3±0.1 | 89.1±0.1 |
| | MADAR-U | 66.4±0.4 | **79.4±0.4** | **84.8±0.1** | **85.8±0.3** | 57.3±0.5 | **76.2±0.2** | **89.8±0.1** | **91.5±0.1** |

## 5.3 CLASS-IL

In this set of experiments with EMBER, we have 11 tasks, where the initial task starts with 50 classes—one for each of 50 malware families—and five classes are added in each subsequent task. The performance of these methods, detailed in Table 2, is measured by average accuracy $\overline{\text{AP}}$ with *None* and *Joint* training baselines at an $\overline{\text{AP}}$ of $26.5 \pm 0.2$ and $86.5 \pm 0.4$, respectively. For a very low budget of 100 samples, MADAR methods greatly outperform GRS, with MADAR-R getting 16% higher $\overline{\text{AP}}$. For more reasonable budgets, however, the uniform variant MADAR-U offers the best performance. For example, with a 10K budget, MADAR-U yields at least 84.8% $\overline{\text{AP}}$, which is better than GRS at 83.5% $\overline{\text{AP}}$. They also fare far better than all prior works, with ER, AGEM, GR, RtF, and BI-R below 30%, TAMiL at 38.2%, and iCaRL at only 64.6%. These poor results for the prior methods are in line with other findings in the malware domain (Rahman et al., 2022). For a budget of 20K, MADAR-U reaches $85.8 \pm 0.3$, nearly as good as the Joint baseline that uses a maximum budget over 150 times larger.

We have 11 tasks for the Class-IL setting of AZ-Class. The summary results of all the experiments are shown in Table 2 and benchmarked against *None* and *Joint* with $\overline{\text{AP}}$ of $26.4\pm0.2$ and $94.2\pm0.1$, respectively. As we can from Table 2 that, among TAMiL, iCaRL, ER, AGEM, GR, RtF, and BI-R, iCaRL outperforms in most of the budget configurations. Therefore, we discuss the results of MADAR in comparison with iCaRL. For a low budget of 100, iCaRL and GRS get less than 44%, while all MADAR methods achieve over 57%. As budgets increase, all methods improve, with MADAR-U offering the best results at every budget from 1K to 20K. At 20K, it reaches $91.5\pm0.1\%$, which is 1.4% higher than GRS and 6.9% higher than iCaRL.

In summary, our experiments clearly demonstrate the effectiveness of MADAR's diversity-aware replay techniques in Class-IL for both EMBER and AZ datasets. Additionally, while GRS shows significant improvement with an increased budget, the uniform variants of MADAR are more effective at every budget level. MADAR significantly improves performance in malware classification by mitigating catastrophic forgetting, and they do so using fewer resources.

## 5.4 TASK-IL

In this set of experiments with EMBER, we have 20 tasks with 5 new classes in each task. Table 3 shows a summarized view of this set of experiments, where the performances are presented as the average accuracy over all tasks ($\overline{\text{AP}}$). Note that Task-IL is considered the easiest scenario of continual learning (van de Ven et al., 2022; 2020). The *None* and *Joint* methods, which are the informal lower and upper bounds of this configuration, attain $\overline{AP}$ of 74.6% and $\overline{AP}$ of 97.03%, respectively.

As we can see from Table 3, ER outperforms TAMiL, A-GEM, GR, RtF, and BI-R in all budget configurations and outperforms GRS for few configurations. MADAR, on the other hand, outperforms all the prior methods significantly in lower budget constraints ($100$–$1K$). For instance, MADAR-U reaches $\overline{\text{AP}}$ of 93.9% with only 1K replay samples, compared with 93.6% for GRS. The performance gap among MADAR, ER, and GRS gets closer as the budget increases; however, MADAR variants continue to either outperform or perform on par with other techniques. In particular, the

Table 3: **Task-IL**: Global average accuracies.

| Group | Method | EMBER | | | | AZ | | | |
|---|---|---|---|---|---|---|---|---|---|
| | | Budget | | | | Budget | | | |
| | | 100 | 1K | 10K | 20K | 100 | 1K | 10K | 20K |
| Baselines | Joint | 97.0±0.3 | | | | 98.8±0.2 | | | |
| | None | 74.6±0.7 | | | | 74.5±0.2 | | | |
| Prior Work | TAMiL | 72.8±0.1 | 86.9±0.2 | 90.3±0.1 | 94.2±0.7 | 80.5±0.4 | 91.5±0.2 | 93.5±0.1 | 94.8±0.2 |
| | ER | 67.4±0.3 | 89.5±0.5 | 94.8±0.2 | 95.4±0.1 | 83.6±0.2 | 92.3±0.3 | 96.2±0.1 | 97.5±0.2 |
| | AGEM | 79.6±0.2 | 83.8±0.4 | 86.1±0.2 | 89.3±0.1 | 76.7±0.5 | 85.3±0.1 | 86.7±0.2 | 91.3±0.3 |
| | GR | 79.8±0.3 | | | | 75.6±0.2 | | | |
| | RtF | 77.8±0.2 | | | | 74.2±0.3 | | | |
| | BI-R | 87.2±0.3 | | | | 85.4±0.2 | | | |
| | GRS | 86.9±0.3 | **93.6±0.3** | 94.7±0.3 | 95.0±0.1 | 85.2±0.1 | 90.8±0.1 | 93.5±0.1 | 95.2±0.1 |
| Ours | MADAR-R | 92.1±0.2 | 93.8±0.2 | 94.8±0.2 | **95.6±0.1** | 86.0±0.3 | 92.4±0.1 | 96.7±0.1 | 97.9±0.2 |
| | MADAR-U | **93.4±0.2** | **93.9±0.3** | **95.6±0.1** | **95.8±0.2** | **88.1±0.3** | **94.5±0.3** | **98.1±0.1** | **98.7±0.1** |

MADAR-U variant of MADAR outperforms all the other techniques and attains $\overline{\textbf{AP}}$ of 95.8% with a 20K replay budget, which is close to joint level performance.

Task-IL for AZ contains 20 tasks, each with 5 non-overlapping classes. Our results are shown in Table 3, compared against the *None* and *Joint* benchmarks, with $\overline{AP}$ of 74.5% and 98.8%, respectively. As with EMBER, ER outperforms TAMiL, AGEM, GR, RtF, BI-R, and GRS for most budgets, so we use it for comparison. For a low budget of 100, MADAR-U achieves an $\overline{AP}$ of 88.1%, 4.5% higher than that of ER. For a higher budget of 20K, MADAR-U attains an $\overline{AP}$ of 98.7%, which is 1.2% higher than that of ER and very close to the joint level performance of 98.8%.

## 5.5 DISCUSSION

Our results show that MADAR outperforms previous methods in Domain-IL, Class-IL, and Task-IL scenarios on both the EMBER and AZ datasets, demonstrating the effectiveness of diversity-aware replay in maintaining stability in continual malware classification. In all scenarios, MADAR outperforms prior replay-based methods that were designed for computer vision tasks. These prior approaches generally performed quite poorly in the malware domain, echoing findings of Rahman et al. (2022). For larger budgets, MADAR's performance approaches the *Joint* baseline while using a much smaller training budget, e.g., as little as 3% of the cost.

The Ratio variant performed better in Domain-IL, while the Uniform variant worked well in Class-IL and Task-IL. Intuitively, this makes sense, as ratio budgeting captures the contributions of each family to the overall malware distribution for binary classification in the Domain-IL setting. Additionally, since there are many small families in the Domain-IL datasets, uniformly sampling from them consumes budget while offering little improvement in malware coverage. In contrast, Class-IL and Task-IL require classification across families, where uniform budgeting ensures class balance and comprehensive coverage.

GRS performs well, often close to MADAR's performance, especially at higher budgets in Domain-IL. GRS provides an unbiased estimate of the underlying distribution, which makes it a strong baseline. MADAR is particularly effective in Class-IL, Task-IL, and lower-budget Domain-IL, while GRS generally performs as well as MADAR in higher-budget Domain-IL. We hypothesize that diversity is more important when the number of samples per class is limited.

## 6 CONCLUSION

In this paper, we propose MADAR, a framework for diversity-aware replay in continual learning specially designed for the challenging setting of malware classification. Our comprehensive evaluation across Domain-IL, Class-IL, and Task-IL scenarios against Windows executable (EMBER) and Android application (AZ) datasets demonstrates that diversity-aware sampling is helpful for effective CL in malware classification. As malware and goodware continue to evolve, we hope these insights steer continual learning towards strategic, resource-efficient methods, ensuring model effectiveness amid the constantly shifting landscape of cybersecurity threats.

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

# A    MADAR IN CLASS-IL AND TASK-IL

# B    EXPERIMENTAL DETAILS

## B.1    DATASET DETAILS

### B.1.1    WINDOWS PE FILES

For our experiments, we chose the EMBER 2018 version, containing features from one million Windows Portable Executable (PE) files, predominantly scanned in 2018.[1] The dataset comprises 400K goodware and 350K malware, with the rest labeled as unknown. EMBER provides a diverse array of 2,381 hand-crafted features, covering general file information, header data, import/export functions, and section details. Notably, these features capture strong semantic concepts that have a limited space of feasible settings, outside of which the executable does not actually run.

---

[1]`https://github.com/elastic/ember`

---

**Algorithm 2:** MADAR in Class-IL and Task-IL

---

**Input** : $c$ – Task number, $X_c, Y_c$ – Malware samples and their family labels, $\mathcal{P}$ – Malware data pool, $\beta$ – Memory budget, $\Omega$ – Split of anomalous/similar samples, $\xi$ – Ratio budgeting, $\Psi$ – Uniform budgeting

1   **init** $\mathcal{P}$;    **init** $\mathcal{D} \leftarrow \{M_f : M_c\}$;

2   **if** $c = 0$ **then**

3     $\mathcal{P} \leftarrow X_c, Y_c; X_{mal} \leftarrow \mathcal{P}; \nabla\mathcal{D} \leftarrow X_{mal}; X_{train}, Y_{train} \leftarrow X_c, Y_c;$

4   **else**

5     $X_{mal} \leftarrow \mathcal{P}; \beta_A, \beta_S \leftarrow \beta \cdot \Omega;$

6     **if** $\Psi$ **then**

7      $\mathcal{NF} \leftarrow \mathcal{D}; \mathcal{B}_\mathcal{F} \leftarrow \beta/\mathcal{NF};$

8     $R_{mal} \leftarrow [\,]$; **for** $X_f \subseteq X_{mal}$ **do**

9      $\mathcal{F}_{\mathcal{MC}} \leftarrow X_f$; **if** $\xi$ **then**

10       $\mathcal{MC} \leftarrow \mathcal{D}; \mathcal{B}_\mathcal{F} \leftarrow (\mathcal{F}_{\mathcal{MC}}/\mathcal{MC}) \cdot \beta;$

11      **if** $\mathcal{F}_{\mathcal{MC}} \leq \mathcal{B}_\mathcal{F}$ **then**

12       $R_{mal}.\texttt{append}(X_f);$

13      **else**

14       $(A_f, S_f) \leftarrow \texttt{IF}(X_f, \beta_A, \beta_S); R_{mal}.\texttt{append}(A_f, S_f);$

15     $X_{replay} \leftarrow R_{mal}; Y_{replay} \leftarrow ([1] \times \texttt{len}(R_{mal}));$

16     $X_{train} \leftarrow \texttt{concat}(X_c, X_{replay}); Y_{train} \leftarrow \texttt{concat}(Y_c, Y_{replay});$

17     $\mathcal{P}.\texttt{append}(X_c, Y_c); \nabla\mathcal{D} \leftarrow X_{mal};$

18   **return** $(X_{train}, Y_{train})$

---

In our Class-IL experiments, we focused on 2018 malware samples from 2,900 families. After filtering out families with fewer than 400 samples, we narrowed the remaining samples down to the top 100 families, leaving 337,035 samples for analysis. For Domain-IL, we included both goodware and malware from the entire year of 2018 for binary classification, excluding unknown samples.

### B.1.2 ANDROID APK FILES

Additionally, we collected two datasets from AndroZoo (Allix et al., 2016) (AZ) for our experiments: AZ-Domain for Domain-IL and AZ-Class for Class-IL and Task-IL. These datasets contain Android APK files, and both use a 9:1 ratio of goodware to malware to reflect the real-world class imbalance. Following the practice of prior work (Xu et al., 2019), the malware samples are selected with a VirusTotal detection count of $>= 4$. The AZ-Domain dataset includes 80,690 malware and 677,756 goodware samples from 2008 to 2016. We divided the AZ-Domain dataset into non-overlapping yearly training and testing sets. The AZ-Class dataset consists of 285,582 samples from 100 Android malware families, each with at least 200 samples.

We extracted Drebin features (Arp et al., 2014) from the apps for both datasets. These features cover various aspects of app behavior, including hardware access, permissions, app component names, filtered intents, restricted API calls, used permissions, suspicious API calls, and network addresses. Again, we note that these capture strong semantic concepts from the operation of the application. The training sets of AZ-Domain and AZ-Class have 3,858,791 and 1,067,550 features, respectively. We processed the test datasets to match the training feature sets and reduced dimensionality by filtering features with low variance ($< 0.001$) using `scikit-learn`'s `VarianceThreshold`. This resulted in final feature dimensions of 1,789 for AZ-Domain and 2,439 for AZ-Class, respectively.

### B.2 MODEL SELECTION AND IMPLEMENTATION

We use a multi-layer perceptron (MLP) model for malware classification, similar to the model used by Rahman et al. (Rahman et al., 2022), for experiments with the EMBER dataset. For the AZ dataset, we developed a new MLP model with five fully-connected layers, quite similar to the MLP used for EMBER. This model uses the Adam optimizer with a learning rate of $0.001$, and batch normalization and dropout for regularization.

The implementation of the output layer varies among Domain-IL, Class-IL, and Task-IL scenarios. Domain-IL operates as a series of binary classification tasks over 12 months for EMBER, and over 9 years for the AZ dataset, with two output units in each case: malicious and benign. In Class-IL, the output layer comprises units – one for each class. Output units are active only if they correspond to classes that have been seen by that point in the experiment. Class-IL begins with an initial set of 50 classes in the first task and progressively adds five more classes in each of the remaining 10 tasks for both EMBER and AZ datasets. In Task-IL, only the output units of the classes in the current task are active. Both the EMBER and AZ-Class datasets divide the classes equally into 20 tasks, with each task containing five classes.

### B.3 BASELINES AND METRIC

**Global Reservoir Sampling (GRS).** GRS simply selects samples at random from a global stored data pool (Vitter, 1985; Zhang & Sutton, 2017). Given a memory budget $\beta$, GRS randomly picks $\beta$ samples from a data pool $\mathcal{P}$, with each incremental learning task contributing to the pool. If $\beta \geq \mathcal{P}$, GRS selects all the available samples in $\mathcal{P}$. Rahman et al. (Rahman et al., 2022) investigated GRS – which they refer to as Partial Joint Replay – only for Domain-IL scenario of EMBER dataset. In this work, we present a deeper investigation of GRS in both Domain-IL and Class-IL scenarios with both EMBER and AZ datasets.

**Global average accuracy ($\overline{AP} \in [0, 100]\%$).** To maintain consistency with prior work, we present results using *global average accuracy* as the primary metric for our evaluations (Rahman et al., 2022; van de Ven et al., 2020; Rebuffi et al., 2017). Note that we conducted a subset of evaluations using other metrics, such as F1 score, precision, and recall, which are not included in this paper. The conclusions remain unchanged for all of these metrics.

Let $P_{i,j}$ be the accuracy of the model on the test set of task $T_j$, $j \leq i$, after continually training the model on tasks 1 to $i$. For $N$ total tasks, the global average accuracy $\overline{AP}$ over all tasks is computed as:

$$\overline{AP} = \frac{1}{N} \sum_{i=1}^{N} \left( \frac{1}{i} \sum_{j=1}^{i} P_{i,j} \right) * 100\% \tag{1}$$

### B.4 TRAINING AND EVALUATION PROTOCOL

A continual learning (CL) model is sequentially trained to learn tasks from $t_1, t_2, ..., t_T$, each with its distinct data distribution $p(x, y|t_i)$. The goal is to adapt to new tasks without forgetting the old ones. CL training involves three sets of parameters: shared parameters ($\theta_s$) across all tasks, old task-specific parameters ($\theta_0$), and new task parameters ($\theta_n$) (Li & Hoiem, 2017). The *Joint* training benchmark trains the model with all the available training samples up to the current task and optimizes all these parameters simultaneously; however, it incurs incremental storage and training costs. In contrast, CL training strives to optimize and update $\theta_s$ and $\theta_n$, while maintaining $\theta_0$ in a relatively fixed state for each new task $t_n$. However, updating any of the shared weights $\theta_s$ risks confusing the classifier when faced with older data, as those classification decisions depend not only on $\theta_0$ but also on $\theta_s$. CL training typically boasts significantly faster speeds and far less storage requirements than *Joint* training, thus permitting more frequent model retraining to adapt to evolving data distributions or other requirements.

In our evaluations, we use a non-overlapping hold-out set corresponding to each task. For example, the AZ-Domain dataset contains 9 years of training samples from 2008 to 2016, resulting in 9 hold-out sets, one for each year. A CL model is evaluated on all the hold-out sets up to the current task; formally, the model is evaluated on tasks $t_i$ to $t_T$, for $1 \leq i \leq T$, after it been trained on the current task $t_T$.

In this work, each set of experiments is performed around 10-15 times with different random parameter initializations. We use PyTorch on a CentOS-7 machine with an Intel Xeon processor, 40 CPU cores, 128GB RAM, and four GeForce RTX 2080Ti GPUs, each with 12GB memory.

# C ADDITIONAL RESULTS

## C.1 DOMAIL-IL

Table 4: Additional Results: Summary of EMBER and AZ Domain-IL Experiments.

| Group | Method | EMBER | | | AZ | | |
|-------|--------|-------|---|---|-----|---|---|
| | | Budget | | | Budget | | |
| | | 10K | 50K | 300K | 10K | 50K | 300K |
| Prior Work | ER | 73.5±0.5 | 70.5±0.3 | 70.0±0.1 | 40.1±0.1 | 41.1±0.2 | 45.9±0.1 |
| | AGEM | 73.6±0.2 | 70.4±0.3 | 70.0±0.1 | 47.4±0.2 | 49.2±0.2 | 54.8±0.4 |
| | GR | | 93.1±0.2 | | | 93.3±0.4 | |
| | GRS | 94.1±1.3 | **95.3±0.2** | 95.8±0.6 | **96.4±0.1** | **96.9±0.1** | **97.2±0.1** |
| Ours | MADAR-R | **94.7±0.1** | **95.4±0.1** | **96.1±0.1** | **96.6±0.1** | **96.9±0.1** | 97.0±0.1 |
| | MADAR-U | 94.0±0.2 | 95.1±0.1 | 95.7±0.1 | 95.5±0.1 | 95.2±0.2 | 95.8±0.2 |

At a 10K budget in EMBER, MADAR-R achieves the highest accuracy at 94.7%, lightly higher than GRS at 94.1%. MADAR outperforms prior works which score significantly lower. At the 300K budget, MADAR-R reaches 96.1%, while GRS follows closely at 95.8%.

In the AZ-Domain, MADAR-R reaches 96.6% at the 10K budget, slightly better than GRS at 96.4%. At 300K, GRS performs marginally better with 97.2%, compared to 97.0% for MADAR-R. Both methods outperform prior work, which show much lower accuracy at all budget levels. These results demonstrate that MADAR-R performs well across budgets, with GRS having a slight edge at higher budgets.

## C.2 CLASS-IL

Table 5: Additional Results: Summary of EMBER and AZ Class-IL Experiments.

| Group | Method | EMBER | | | AZ | | |
|-------|--------|-------|---|---|-----|---|---|
| | | Budget | | | Budget | | |
| | | 500 | 5K | 15K | 500 | 5K | 15K |
| Prior Work | TAMiL | 33.1±0.2 | 36.7±0.1 | 37.2±0.2 | 55.2±0.3 | 60.8±0.2 | 65.3±0.5 |
| | iCaRL | 58.7±0.7 | 63.9±1.2 | 65.5±1.0 | 54.9±1.0 | 77.2±0.4 | 83.4±0.5 |
| | ER | 27.8±0.1 | 27.9±0.1 | 28.0±0.1 | 58.3±0.6 | 59.2±0.8 | 63.1±0.5 |
| | AGEM | 27.4±0.1 | 28.5±0.1 | 28.3±0.1 | 28.0±1.4 | 28.0±0.6 | 29.8±2.6 |
| | GR | | 26.8±0.2 | | | 22.7±0.3 | |
| | GRS | 70.3±0.5 | 82.0±0.2 | 84.3±0.3 | 62.9±0.8 | 83.0±0.3 | 88.2±0.2 |
| Ours | MADAR-R | 73.6±0.2 | 81.5±0.2 | 83.8±0.2 | 67.8±0.9 | 82.9±0.2 | 88.2±0.2 |
| | MADAR-U | **76.5±0.2** | **83.8±0.2** | **85.5±0.1** | **70.4±0.4** | **86.8±0.1** | **91.0±0.1** |

For EMBER Class-IL experiments, at a budget of 500, MADAR-U achieves the highest accuracy of 76.5%, outperforming GRS at 70.3%. As the budget increases to 15K, MADAR-U maintains its lead with an accuracy of 85.5%, slightly ahead of GRS at 84.3%. Prior methods like iCaRL and TAMiL show lower performance, with iCaRL reaching 65.5% and TAMiL only achieving 37.2% at the highest budget. This demonstrates the effectiveness of MADAR methods, particularly MADAR-U, across all budget levels.

For the AZ dataset, MADAR-U also leads with 70.4% accuracy at the 500-sample budget, outperforming GRS at 62.9%. At the 15K budget, MADAR-U reaches 91.0%, compared to GRS which achieves 88.2%. In contrast, prior methods like iCaRL and TAMiL show much lower performance, with TAMiL yielding at 65.3% and iCaRL at 83.4%. These results highlight the consistent effectiveness of MADAR-U across both EMBER and AZ domains.

## C.3 TASK-IL

In the EMBER Task-IL experiments, MADAR-U shows the highest accuracy at all budget levels with 93.7% at a budget of 500 sample and reaching 95.7% at the 15K budget. MADAR-R also performs well, with an accuracy of 92.3% at 500 budget and matching MADAR-U at 15K with

Table 6: Additional Results: Summary of EMBER and AZ Task-IL Experiments.

| Group | Method | EMBER Budget | | | AZ Budget | | |
|---|---|---|---|---|---|---|---|
| | | 500 | 5K | 15K | 500 | 5K | 15K |
| Prior Work | TAMiL | 81.5±0.3 | 88.1±0.3 | 93.2±0.3 | 85.3±0.6 | 92.1±0.1 | 94.0±0.2 |
| | ER | 84.9±0.2 | 93.9±0.2 | 95.2±0.1 | 90.2±0.1 | 95.6±0.1 | 97.1±0.2 |
| | AGEM | 81.7±0.2 | 84.9±0.2 | 88.9±0.2 | 82.8±0.2 | 85.6±0.2 | 89.2±0.2 |
| | GR | | 79.8±0.3 | | | 75.6±0.2 | |
| | GRS | 87.4±0.3 | 94.4±0.2 | 94.9±0.1 | 89.2±0.2 | 91.6±0.2 | 93.9±0.1 |
| Ours | MADAR-R | 92.3±0.9 | 94.2±0.1 | **95.7±0.2** | 90.3±0.2 | 95.8±0.2 | 97.1±0.1 |
| | MADAR-U | **93.7±0.3** | **94.8±0.2** | **95.7±0.1** | **92.9±0.2** | **97.2±0.2** | **98.2±0.1** |

95.7%. Both methods outperform previous approaches, including ER and GRS, which reach 95.2% and 94.9% at the 15K budget, respectively. These results demonstrate that MADAR methods, particularly MADAR-U, are highly effective in the Task-IL setting.

For the AZ dataset, MADAR-U continues to lead, starting at 92.9% accuracy at the 500 budget and reaching 98.2% at 15K. MADAR-R follows with 90.3% at 500 budget and 97.1% at 15K. In comparison, prior methods such as ER and GRS achieve 97.1% and 93.9%, respectively, at the 15K budget. These results show a consistent effectiveness of MADAR-U across all budget levels in the AZ dataset.

