# OpenReview forum: "MADAR: Efficient Continual Learning for Malware Analysis with Diversity-Aware Replay"
_ICLR.cc/2025/Conference — Submitted to ICLR 2025_

### Official Review · Reviewer_ZtFk · 2024-10-30

**Soundness:** 3
**Presentation:** 3
**Contribution:** 3
**Rating:** 5
**Confidence:** 4

**Summary:**

This paper addresses a major issue in malware classification -- that malware samples evolve rapidly and machine learning classifiers must be retrained continually to keep up with the evolving landscape. However, continual learning techniques developed for computer vision tasks do not work well for malware because malware families have huge imbalance and diversity between and within classes. This paper proposes MADAR, which does a diversity aware sampling of malware families to be used during retraining in CL setups. The method shows impressive performance gains in class-IL and task-IL scenarios.

**Strengths:**

- Addresses important issues of concept drift and catastrophic forgetting in the malware domain
- Proposed approach is simple enough and reduces the cost of CL
- Experiments with several baselines show improvements in performance
- Well-written paper, easy to understand

**Weaknesses:**

- Design decisions are often not motivated enough/missing rationale (see questions)
- Tables I-III seem to report accuracy instead of balanced accuracy even though classes are imbalanced, leading to unclear evaluation
- For domain-IL, the improvements seem marginal (even the entirety of CL seems unnecessary when looking at baseline numbers)

**Questions:**

- Page 4 describes how tasks are created. "As our datasets do not possess naturally defined tasks, we partition our dataset into tasks comprising an equal number of independent and non-overlapping classes to act as a proxy to new behaviors" -> How representative is an equal class size in terms of generalization to open-world settings? Some tasks are obviously common than others, so wouldn't they require a corresponding sampling ratio?

- The result comparison between uniform and ratio budgeting seems interesting. However, along the same lines as the previous question, uniform sampling does not take into account that class sizes (malware families or behaviors) are not equally distributed in open-world settings. So, the uniform sampling seems to create an artificial class distribution for the classifier. Combined with the evaluation metric (average accuracy across many tasks), how do the authors minimize the risk of overfitting?

- In terms of evaluation metrics (Tables I-III), please explain the rationale for using accuracy instead of metrics that take into account class imbalance (e.g., balanced accuracy, precision, recall). It would also be interesting to see a task-wise (month/year-wise) distribution of results so a true measure of performance can be gleaned.

- For isolation forests, the ratio C_r is chosen to be 0.1. How is this value chosen and is it representative to real-world malware distributions?

- Why was a value of 0.5 chosen as the ratio of representative and anomalous samples? Wouldn't it seem more intuitive to have more representative samples than anomalous ones?

- There is a huge difference in results for Domain-IL vs. Task-IL or Class-IL. Please explain what causes this difference, i.e., is it merely binary vs. multi-class setting that is causing the difference or are there inherent differences in task structures?

---

### Official Review · Reviewer_nTkb · 2024-11-04

**Soundness:** 4
**Presentation:** 3
**Contribution:** 2
**Rating:** 5
**Confidence:** 4

**Summary:**

The paper presents MADAR (Malware Analysis with Diversity-Aware Replay), a continual learning framework for malware classification that addresses catastrophic forgetting by selectively replaying diverse malware samples. Unlike existing techniques that struggle with malware data, MADAR uses a diversity-aware strategy, preserving both common and rare (anomalous) samples within each malware family through Isolation Forest-based sampling. This approach enables MADAR to efficiently maintain high accuracy with minimal memory requirements. Tested on both Windows (EMBER) and Android (AndroZoo) malware datasets across three continual learning scenarios, MADAR significantly outperforms traditional replay methods, offering a resource-efficient solution for real-world malware detection in constantly evolving threat landscapes.

I have enjoyed reading the paper. It was very easy to follow. I appreciate the authors for their efforts.

**Strengths:**

1. Identifying the challenges of continual learning in the malware domain, and addressing them. It is good to see that recent works in the malware domain are adapting well-known ML techniques.

2. Proposing two different budgeting (random and uniform) for three different scenarios, which are simple but efficient.

3. Inclusion of both windows PE and android APK datasets in the evaluation.

**Weaknesses:**

1. While the contribution is smart, it is not enough in terms of the evaluation result, e.g., the proposed method has achieved marginal improvement from the existing method GRS. If random sampling in GRS is on par with the proposed method, then why should we use MADAR?

2. The 'Related Work' section is not comprehensive enough. There have been many works on the concept drift of malware, and difficulties of ML in the security domain, such as [1, 2]. Moreover, like CL, there have been other recent works in the malware domain that were also adapted from vision, such as [4, 5].

3. No comparison against the method of Chen et. al. [3] was shown. As this is one of the pioneering works of CL in the malware domain, I believe the authors should compare their work against Chen's method.

4. No strategy for sampling goodware samples was proposed.

5. *"We found empirically that a balanced split $(\alpha = 0.5)$ between representative and anomalous samples provides optimal performance."* It would be better to show an ablation experiment or analysis of this.

6. If the AZ dataset has a 9:1 benign to malware ratio, then it is counter-intuitive to use it to show that MADAR is good for continual learning of malware, when it might be the case that the most of performance boost is coming from the goodware. I would highly recommend the author to randomly pick a subset of goodware to make the ratio 1:1, and then run the evaluation.

References:

[1] Dos and Don’ts of Machine Learning in Computer Security

[2] Demystifying Behavior-Based Malware Detection at Endpoints

[3] Continuous Learning for Android Malware Detection

[4] RS-Del: Edit Distance Robustness Certificates for Sequence Classifiers via Randomized Deletion

[5] DRSM: DE-RANDOMIZED SMOOTHING ON MALWARE CLASSIFIER PROVIDING CERTIFIED ROBUSTNESS

**Questions:**

1. How are the replay goodwares sampled for training and evaluation? Were they just randomly sampled?

2. Why $\alpha$ was chosen as $0.5$ for representative and anomalous samples?

3. I can see in the subsection 5.2 that the Joint baseline used 670K samples. If that is right, can you mention this number in the table or at least in the subsection 5.1 where you discuss the dataset?

---

### Official Review · Reviewer_e5cH · 2024-11-05

**Soundness:** 2
**Presentation:** 2
**Contribution:** 2
**Rating:** 3
**Confidence:** 3

**Summary:**

The paper introduces MADAR, a framework for continual learning in malware analysis by using diversity-aware replay to select representative and anomalous samples. Through the evaluations on Windows and Android malware datasets, the authors illustrate the effectiveness of MADAR. Nevertheless, the novelty of this paper is not high, since the solution is ordinary and the addressed problems have been studied by many works.

**Strengths:**

Promising topic, Malware detection is a representative task, and considering continuous learning to alleviate the overhead caused by retraining is a problem worth studying.
Using representative malware datasets involving large-scale malware samples and features.

**Weaknesses:**

The novelty of this paper is not high, since the solution is ordinary and the addressed problems have been studied by many works. Specifically, the Isolation Forest is only used in the methods to split the samples, yet the model improvements are not enough. Also, the researched question needs to be further clarified, thus indicating the challenges and the parts that have not been done by others.

**Questions:**

What are the differences and advantages of the proposed scheme compared to the current methods for known and unknown malware detection?
In the security community, a series of arts have been proposed previously involving class-incremental learning and unknown detection (including isolation forests), which I think should be discussed or compared. Some of the literature is as follows.

[1] FARE: Enabling Fine-grained Attack Categorization under Low-quality Labeled Data. NDSS 2021.
https://www.ndss-symposium.org/ndss-paper/fare-enabling-fine-grained-attack-categorization-under-low-quality-labeled-data/

[2] FOSS: Towards Fine-Grained Unknown Class Detection Against the Open-Set Attack Spectrum With Variable Legitimate Traffic. TNET 2024. https://ieeexplore.ieee.org/abstract/document/10638516/

[3] Detecting unknown encrypted malicious traffic in real time via flow interaction graph analysis. NDSS 2023. https://arxiv.org/abs/2301.13686

[4] I 2 RNN: An Incremental and Interpretable Recurrent Neural Network for Encrypted Traffic Classification. TDSC 2024. https://ieeexplore.ieee.org/abstract/document/10056861

[5] Random partitioning forest for point-wise and collective anomaly detection—Application to network intrusion detection. TIFS 2021. https://ieeexplore.ieee.org/abstract/document/9319404/

**Details Of Ethics Concerns:**

There are no ethical issues with this paper.

---

### Official Review · Reviewer_mRB6 · 2024-11-06

**Soundness:** 2
**Presentation:** 3
**Contribution:** 1
**Rating:** 3
**Confidence:** 3

**Summary:**

This paper proposes a continual learning framework for malware analysis. The key idea is to mix the samples from previous tasks (i.e., replay) and the new samples from the new task with an emphasis on the data diversity. The evaluation is based on EMBER and AZ datasets and it shows that MADAR significantly outperforms prior work.

**Strengths:**

- The presentation of this paper is good. The introduction to the background related to continual learning and incremental learning is very clear.

- Mixing the previous experience with the new samples to address the challenge of catastrophic forgetting sounds reasonable.

**Weaknesses:**

- The novelty of this work is limited. The replay technique in continual learning has been well established as evidenced in the related work section. Besides this, the contribution of MADAR is limited and there is no theoretical guarantee regarding the impact of this technique towards the final performance.

- Since there is only one work that studied CL in the malware domain, whether it is a real challenge in the malware domain comes into doubt. It would be great if the authors could provide further explanation about why it is a real challenge in the security community. Otherwise, the problem to solve seems artificial.

- The evaluation is limited. First, the selected datasets are not sufficient. I am wondering if it is possible to do experiments on the APIGraph Dataset [1]. Second, the compared baselines are relatively weak and not state-of-the-art. Given the submitted venue is a ML conference, I would expect the authors to include the SOTA incremental learning methods for comparison. Furthermore, the malware detection task has been long studied in the security community. It would be great if the authors could provide empirical results of previous methods published in top-tier security conferences.

- The reproducibility of this work is limited. For example, I even cannot determine which network structure I should use after reading the whole paper. Additionally, it would be better to include some details of the selected datasets in the appendix.

Reference:

[1] X. Zhang, Y. Zhang, M. Zhong, D. Ding, Y. Cao, Y. Zhang, M. Zhang, and M. Yang. Enhancing state-of-the-art classifiers with API semantics to detect evolved Android malware. In Proceedings of the 2020 ACM SIGSAC Conference on Computer and Communications Security, pages 757–770, 2020.

**Questions:**

- How to obtain the labels for the IF method to separate the anomalous data points from the rest?

- How to determine $\gamma$ in practice? How to support that your claim "Our Android malware (AZ) datasets, for example, have a 9:1 ratio of goodware to malware, so we use γ = 0.9" is reasonable and correct?

- How to determine the value of $\alpha$? Is $\alpha=0.5$ a good value in general? Any ablation study would be appreciated.

**Details Of Ethics Concerns:**

There already exists one archived PhD dissertation that introduced MADAR (https://repository.rit.edu/theses/11758/). I am not sure if it belongs to self plagiarism or plagiarism given the double-blind nature of the submission.

---

### Meta-Review · Area_Chair_dANH · 2024-12-15

**Metareview:**

Limited Novelty and Contribution: The replay technique used in this work is well-established, and the contribution of MADAR lacks theoretical guarantees. The use of Isolation Forest for sample splitting is insufficiently innovative, and the addressed problem seems artificial without clear evidence of its relevance to the security community.

Evaluation Issues: The evaluation is inadequate due to limited and non-diverse datasets, weak baselines, and a lack of state-of-the-art (SOTA) methods for comparison. Additionally, results show only marginal improvements over existing methods like GRS, raising questions about the practical benefit of MADAR.

Reproducibility and Clarity: The paper lacks critical implementation details, such as the network structure and dataset characteristics, hindering reproducibility. Design decisions and experimental rationale are poorly motivated, and ablation studies or analysis are missing for key claims.

Inadequate Related Work and Comparisons: The "Related Work" section is not comprehensive, overlooking key studies on concept drift, continual learning, and malware analysis. Furthermore, no comparison is made against foundational works like Chen et al., which weakens the contextual positioning of the research.

Imbalanced Data and Experimental Design Flaws: The use of datasets with imbalanced class ratios (e.g., AZ dataset) is problematic. Evaluation metrics, such as accuracy, fail to account for class imbalance, making results unclear and potentially misleading. Suggestions like balancing data ratios and using more rigorous metrics are not addressed.

**Additional Comments On Reviewer Discussion:**

The authors provided no response to reviewers' questions.

---

### Decision · Program_Chairs · 2025-01-22

Reject